# Investigation of the Antihypertrophic and Antifibrotic Effects of Losartan in a Rat Model of Radiation-Induced Heart Disease

**DOI:** 10.3390/ijms222312963

**Published:** 2021-11-30

**Authors:** Mónika Gabriella Kovács, Zsuzsanna Z. A. Kovács, Zoltán Varga, Gergő Szűcs, Marah Freiwan, Katalin Farkas, Bence Kővári, Gábor Cserni, András Kriston, Ferenc Kovács, Péter Horváth, Imre Földesi, Tamás Csont, Zsuzsanna Kahán, Márta Sárközy

**Affiliations:** 1Interdisciplinary Center of Excellence and MEDICS Research Group, Department of Biochemistry, Albert Szent-Györgyi Medical School, University of Szeged, H-6720 Szeged, Hungary; kovacs.monika.gabriella@med.u-szeged.hu (M.G.K.); kovacs.zsuzsanna@med.u-szeged.hu (Z.Z.A.K.); szucs.gergo@med.u-szeged.hu (G.S.); marah.mf.94@gmail.com (M.F.); 2Department of Oncotherapy, Albert Szent-Györgyi Medical School, University of Szeged, H-6720 Szeged, Hungary; varga.zoltan@med.u-szeged.hu (Z.V.); kahan.zsuzsanna@med.u-szeged.hu (Z.K.); 3Department of Laboratory Medicine, Albert Szent-Györgyi Medical School, University of Szeged, H-6720 Szeged, Hungary; lokine.farkas.katalin@med.u-szeged.hu (K.F.); foldesi.imre@med.u-szeged.hu (I.F.); 4Department of Pathology, Albert Szent-Györgyi Medical School, University of Szeged, H-6720 Szeged, Hungary; kovari.bence@med.u-szeged.hu (B.K.); cserni.gabor@med.u-szeged.hu (G.C.); 5Synthetic and Systems Biology Unit, Biological Research Centre, Eötvös Loránd Research Network, H-6726 Szeged, Hungary; kriston.andras@single-cell-technologies.com (A.K.); kovacs.ferenc@single-cell-technologies.com (F.K.); peter.horvath@brc.hu (P.H.); 6Single-Cell Technologies Ltd., H-6726 Szeged, Hungary; 7Institute for Molecular Medicine Finland (FIMM), University of Helsinki, FIN-00014 Helsinki, Finland

**Keywords:** onco-cardiology, radiation-induced heart disease, diastolic dysfunction, left ventricular hypertrophy, fibrosis, heart failure, angiotensin-II receptor blocker (ARB), losartan, chymase, TGF-β/SMAD signaling pathway

## Abstract

Radiation-induced heart disease (RIHD) is a potential late side-effect of thoracic radiotherapy resulting in left ventricular hypertrophy (LVH) and fibrosis due to a complex pathomechanism leading to heart failure. Angiotensin-II receptor blockers (ARBs), including losartan, are frequently used to control heart failure of various etiologies. Preclinical evidence is lacking on the anti-remodeling effects of ARBs in RIHD, while the results of clinical studies are controversial. We aimed at investigating the effects of losartan in a rat model of RIHD. Male Sprague-Dawley rats were studied in three groups: (1) control, (2) radiotherapy (RT) only, (3) RT treated with losartan (per os 10 mg/kg/day), and were followed for 1, 3, or 15 weeks. At 15 weeks post-irradiation, losartan alleviated the echocardiographic and histological signs of LVH and fibrosis and reduced the overexpression of chymase, connective tissue growth factor, and transforming growth factor-beta in the myocardium measured by qPCR; likewise, the level of the SMAD2/3 protein determined by Western blot decreased. In both RT groups, the pro-survival phospho-AKT/AKT and the phospho-ERK1,2/ERK1,2 ratios were increased at week 15. The antiremodeling effects of losartan seem to be associated with the repression of chymase and several elements of the TGF-β/SMAD signaling pathway in our RIHD model.

## 1. Introduction

Cardiovascular diseases and cancer are the leading causes of morbidity and mortality worldwide [1,2]. The most common cancerous diseases are breast and lung cancers in women and men, respectively [2]. Cancer therapy has undergone significant improvement, which led to increased long-term survival rates among cancer patients [3]. About 50% of cancer patients receive radiotherapy (RT), which also has an important role in the treatment of malignancies superposed on the chest wall, such as breast cancer and thoracic malignancies including lung, and esophageal cancers, Hodgkin’s lymphoma, and thymoma [3,4]. While high-energy ionizing radiation (i.e., RT) successfully kills tumor cells, it could have harmful effects on the surrounding healthy tissues [5]. Depending on the RT technique and dose used in thoracic and breast malignancies, the heart can be at risk of being exposed to ionizing radiation resulting in radiogenic sequelae in a dose-dependent manner [6,7]. The syndrome of unwanted cardiovascular side effects of thoracic RT is termed radiation-induced heart disease (RIHD), which is a critical concern in current oncology and cardiology practice [8,9,10].

RIHD is a progressive multifactorial disease that covers a broad spectrum of cardiac pathology [8,9,10]. Its clinical manifestation includes acute and chronic pericarditis, conduction system abnormalities, ischemic heart disease, cardiomyopathy, heart failure with preserved ejection fraction (HFpEF) or reduced ejection fraction (HFrEF), and valvular heart disease [8,9,10]. RT simultaneously causes damage to the cardiac macrovasculature (i.e., coronary arteries) and microvasculature, as well as the myocardium (i.e., diffuse injury), leading to the complex pathomechanism of RIHD [11,12]. However, the precise molecular mechanisms in the progression of RIHD from acute to chronic heart diseases are not clearly understood yet. Evidence suggests that RT-induced direct nitro-oxidative damage of macromolecules, including DNA, proteins, and lipids, initiates the development of RIHD. At this acute phase of RIHD, the elevated nitro-oxidative stress causes injury to the endothelial and other cells, eventually leading to various forms of cell death and acute inflammation [8,12,13]. In the early chronic phase of RIHD, the sublethally damaged surviving cardiomyocytes develop hypertrophy accompanied by endothelial cell proliferation as a compensatory mechanism [8,12,13]. If these compensatory mechanisms are exhausted, chronic inflammatory processes, fibrosis, and endothelial senescence play the primary role in the progression of RIHD [8,12,13]. Several pathomechanisms, including nitro-oxidative stress, cell death, and inflammatory processes, overlap in the acute and chronic phases of RIHD [8]. The injury of the capillaries or coronary arteries disturbs circulation and leads to hypoxia which aggravates tissue damage [8]. These mechanisms seem to activate and potentiate each other leading to a vicious cycle in the progression of RIHD [8]. Unfortunately, therapeutic options for RIHD are currently insufficient. Therefore, understanding the exact molecular mechanisms in the progression of RIHD is essential for developing preventive and therapeutic strategies together with testing drugs that do not interfere with the anti-cancer effects of RT.

Chronic activation of the renin-angiotensin-aldosterone system (RAAS) plays a pivotal role in cardiovascular pathophysiology, including hypertension, cardiac hypertrophy, and heart failure via different systemic and tissue-specific mechanisms such as elevated nitro-oxidative and endoplasmic reticulum stress, inflammation, apoptosis, and fibrosis via transforming growth factor-β (TGF-β) signaling, and the transactivation of various intracellular protein kinases such as ERKs and AKT [14,15]. There is some preclinical evidence that irradiation could upregulate angiotensin-II (AngII) expression in the rat heart [16] and lungs [17] in a dose-dependent manner [18]. Interestingly, preclinical studies evaluating the cardiac effects of the widely used selective AngII type 1 (AT1) receptor blockers in RIHD are lacking in the literature. Only two clinical studies investigated the effects of ARBs in cancer patients treated with thoracic RT, and their results were controversial [19,20]. In the present study, we aimed at investigating the effects of the ARB losartan (widely used in standard heart failure therapy) in a rat model of RIHD.

## 2. Results

### 2.1. Systemic Effects of the Radiotherapy in Our RIHD Model

An overview of the experimental setup is shown in Figure 1. Altogether two animals died in the RT only group (in the 3-week and 15-week subgroups).

At weeks 1, 3, and 15, hearts, lungs, and tibias were isolated, then left and right ventricles were separated, and the organ weights and tibia lengths were measured (Figure 1 and Table 1). Before the treatments, there were no significant differences in the body weights between the groups in each experiment (Table 1). At week 1, there were no significant differences in the tibia length, heart weight, lung weight, and right ventricular weight between the groups. In contrast, body weight was significantly lower in the RT plus losartan group as compared to the control group (Table 1). Moreover, left ventricular weight was significantly reduced in both RT groups irrespective of the addition of losartan treatment as compared to the control group at week 1 (Table 1). At week 3, there was no significant difference in tibia length, heart weight, or right ventricular weight between the groups (Table 1). However, body weight and left ventricular weight were significantly smaller in the RT groups irrespective of losartan treatment compared to the control group at week 3, probably due to the detrimental effects of ionizing radiation in the RT groups (Table 1). Moreover, lung weights were significantly higher both in the RT only and losartan-treated RT groups compared to the control group, suggesting the presence of pulmonary edema at week 3 (Table 1) (Fluid was found in the lungs during the autopsy, data not shown). At week 15, body weight, tibia length, heart weight, and left ventricular weight were significantly smaller in both RT groups, irrespective of losartan treatment, indicating that the irradiated animals had severe developmental retardation in the late chronic phase of RIHD (Table 1). In contrast, right ventricular and lung weights were not significantly different between the groups (Table 1).

Blood counts, hemoglobin concentrations, and hematocrit levels were determined at weeks 1, 3, and 15 to investigate the effects of losartan on the severity of systemic inflammation and the compensatory increase in hemoglobin synthesis associated with radiation-induced lung and heart damage (Figure 1). One week post-RT, white blood cell counts were not significantly higher (7% increase, *p* = 0.646) in the RT only group and significantly increased in the RT plus losartan group as compared to that in the control group (Table 2). At week 3, white blood cell counts were significantly increased in the irradiated groups irrespective of losartan treatment, suggesting the presence of systemic inflammation at that time point (Table 2). No such differences were detected between the groups at week 15 (Table 2). RT might lead to the activation of platelets which are considered important sources of pro-thrombotic agents during the inflammatory process [21]. Platelet counts were not significantly different between the groups at weeks 1 and 15. However, platelet counts showed a non-significant increase (16%, *p* = 0.073) in the RT only group and a significant increase in the losartan-treated RT group as compared to that in the control group at week 3 (Table 2).

At week 1, red blood cell counts were significantly elevated in both RT groups compared to the control group (Table 2). Hemoglobin level showed a trend to an increase (10%, *p* = 0.057) in the RT only group and a significant increase in the RT plus losartan group compared to the control group (Table 2). There was no significant difference in the hematocrit levels between the groups (Table 2). At week 3, there were no significant differences in the red blood cell counts, hemoglobin, or hematocrit levels between the groups (Table 2). In contrast, at week 15, red blood cell counts showed a trend to an increase (10%, *p* = 0.076) in the RT only group, and a significant increase in the RT plus losartan group, while hemoglobin concentrations and hematocrit levels were significantly increased in the RT groups irrespective of losartan treatment compared to the control group (Table 2).

### 2.2. Post-RT Diastolic Dysfunction Was Alleviated by Losartan at Weeks 1 and 3 but Not at Week 15

Transthoracic echocardiography was performed at weeks 1, 3, and 15 to monitor the effects of RT and losartan on cardiac morphology and function (Figure 1). At week 1, the diastolic parameter mitral valve early flow velocity (E) was not significantly different between the groups (Table 3). A sensitive parameter of the diastolic function, the septal mitral annulus velocity (e’), was significantly decreased in the RT only group as compared to the control group, indicating an early-phase diastolic dysfunction (DD) at week 1 (Table 3, Figure 2a). Septal e’ was not significantly different between the control and the losartan-treated RT groups (Table 3, Figure 2a). Another indicator of DD, the E/e’ ratio, was not significantly different between the RT only group and the control group. However, the E/e’ ratio was significantly reduced in the RT plus losartan group as compared to the RT only or control groups at week 1 (Figure 2b).

At week 3, E velocity was not significantly different between the RT only and the control group (Table 3). In contrast, E velocity was significantly lower in the losartan-treated RT group than in RT only group at week 3 (Table 3). At week 15, E velocity was significantly lower in both RT groups irrespective of losartan treatment compared to the control group (Table 3). At weeks 3 and 15, the significantly decreased septal e’ and significantly increased E/septal e’ ratio indicated the presence of DD in the RT only groups compared to the time-matched control groups, respectively (Table 2, Figure 2a,b). Losartan significantly ameliorated these parameters compared to the RT group at week 3 (Table 2, Figure 2a,b). Nonetheless, at week 15, there was a trend to an increase (13.5%, *p* = 0.061) in the septal e’ and a trend to decrease (17%, *p* = 0.16) in the E/septal e’ parameter in the RT plus losartan group compared to the RT only group (Table 2, Figure 2a,b).

Irradiation and cardiac remodeling might lead to heart rate changes [8]. There were no significant differences in the heart rate between the groups at weeks 1 and 3 (Figure 2c). In contrast, the heart rate was significantly decreased in both RT groups irrespective of losartan treatment at week 15 (Figure 2c).

### 2.3. RT-Related Echocardiographic Signs of Left Ventricular Hypertrophy (LVH) Were Alleviated by Losartan at Week 15

At week 1, there was no significant difference in wall thicknesses and left ventricular diameters, and ejection fraction between the RT only and control groups (Table 3, Figure 2d–f). Nevertheless, in the RT only group, the systolic inferior and posterior wall thicknesses increased in a statistically non-significant manner (10%, *p* = 0.14 and 7%, *p* = 0.17, respectively) while the left ventricular end-systolic diameter decreased in comparison with the control group (trend, 15%, *p* = 0.07), supposing the beginning of the development of LVH at week 1 (Table 3, Figure 2d–f). These echocardiographic signs of a mild LVH resulted in significantly increased fractional shortening in the RT only group measured by the Teichholz method in M-mode images at week 1 (Table 3). Interestingly, similar statistically significant changes occurred in the systolic inferior wall thickness and left ventricular end-systolic and end-diastolic diameters in the RT plus losartan group compared to the control group at week 1 (Table 3). The systolic posterior wall thickness showed a trend to increase (13%, *p* = 0.084) in the RT plus losartan group compared to the control group at week 1 (Figure 2e). Notably, there were no significant differences in the echocardiographic morphologic and functional parameters between the RT only and RT plus losartan groups at week 1.

At week 3, inferior and posterior wall thicknesses both in systole and diastole were significantly increased in the RT only group as compared to that in the control group, indicating the development of LVH (Table 3, Figure 2d–f). Accordingly, the left ventricular end-systolic and end-diastolic diameters were significantly smaller, and the fractional shortening was significantly higher in the RT only group as compared to that in the control group at week 3 (Table 3, Figure 2d). There were no significant differences in the systolic and diastolic inferior and systolic posterior wall thicknesses between the control and RT plus losartan groups (Table 3, Figure 2d–f). However, the left ventricular end-diastolic and end-systolic diameters remained significantly smaller, and the fractional shortening was significantly higher in the RT plus losartan group compared to those in the control group at week 3 (Table 3, Figure 2c,d). There were no significant differences in the ejection fraction between the groups at week 3.

At week 15, severe concentric LVH developed in the RT only group with a significant increase in all wall thicknesses and fractional shortening and a marked reduction in the left ventricular end-diastolic and end-systolic diameters compared to the control group (Table 3, Figure 2d–f). After losartan therapy, the systolic and diastolic posterior and diastolic inferior wall thicknesses were significantly reduced, and the left ventricular end-diastolic diameter markedly increased as compared to these values in the RT only group suggesting anti-hypertrophic effects of losartan in our chronic RIHD model at week 15 (Table 3, Figure 2d–f). Moreover, systolic inferior wall thickness and fractional shortening showed a trend of being lower (19%, *p* = 0.068 and 13%, *p* = 0.098, respectively), and the left ventricular end-systolic diameter seemed to be higher (66%, *p* = 0.063) in response to losartan compared to the RT only group at week 15 (Table 3). However, there was no significant difference in the ejection fraction between the groups at week 15.

### 2.4. Cardiomyocyte Hypertrophy and the Overexpression of LVH Markers Were Reduced in the Losartan-Treated Animals 3 and 15 Weeks after RT

Cardiomyocyte cross-sectional areas were measured on hematoxylin–eosin-stained histological slides. Additionally, the left ventricular expressions of cardiac hypertrophy markers were measured by qRT-PCR to resolve the contradiction between the autopsy and echocardiography results on heart size and LVH development (Table 1 and Table 3, Figure 2 and Figure 3).

At week 1, there were no significant differences in the cardiomyocyte cross-sectional areas between the groups, supporting the heart weight results (Figure 3a,b, and Table 1). In contrast, at week 1, the echocardiographic signs showed the initiation of LVH (Table 3 and Figure 2d–f). Therefore, we investigated the expression of the fetal myosin heavy chain β-isoform (i.e., β-MHC or *Myh7*) and the adult myosin heavy chain α-isoform (i.e., α-MHC or *Myh6*), as well as their ratio, in order to use them as molecular markers of LVH (Figure 3c–e). The increased ratio of the β-MHC to the α-MHC is an indicator of the fetal gene reprogramming in LVH in response to tissue hypoxia [22]. At week 1, the expression of α-MHC was significantly decreased, and the expression of β-MHC was tendentiously increased in both RT groups irrespective of losartan treatment (93%, *p* = 0.061 and 80%, *p* = 0.067, respectively, Figure 3c,d). Accordingly, as a compensatory mechanism to cardiac tissue damage due to RT, the β-MHC to α-MHC (*Myh7/Myh6*) ratio was significantly increased in both RT groups compared to the control group, supporting the echocardiographic signs of the initiation of compensatory LVH at week 1 (Table 2).

At week 3, the cardiomyocyte cross-sectional area showed a trend toward a decrease (10%, *p* = 0.08) in the RT only group compared to the control group, paralleling the left ventricular weight results (Figure 3a,b, Table 1). Notably, there were no significant differences in the cardiomyocyte cross-sectional areas between the control and losartan-treated RT groups, supposing the protective effects of losartan in our RIHD model at week 3 (Figure 3a,b). At the molecular level, there was no significant difference in the expression of α-MHC between the groups at week 3 (Figure 3c). In contrast, the β-MHC expression and the β-MHC to α-MHC ratio were significantly higher in the RT only group compared to the control group, supporting the echocardiographic signs of a mild LVH in our RIHD model at week 3 (Figure 3d,e, Table 1). There were no significant differences in the β-MHC expression and the β-MHC to α-MHC ratio between the losartan-treated RT and control groups, similar to the echocardiographic and histologic results in our RIHD model at week 3 (Figure 3d,e, Table 1).

At week 15, the cardiomyocyte cross-sectional area was significantly higher in the RT only group compared to the control group, indicating the development of hypertrophy in the surviving cardiomyocytes as a compensatory mechanism despite the smaller heart size in our RIHD model (Figure 3a,b, Table 1). Moreover, there were no significant differences in the cardiomyocyte cross-sectional areas between the losartan-treated RT and control groups, similarly to the echocardiographic results on the anti-hypertrophic effects of losartan at week 15 (Figure 3a,b, Table 1). The α-MHC expressions were markedly decreased, and the β-MHC expressions with the β-MHC to α-MHC ratios were significantly increased in the RT only group as compared to that in the control group, indicating the development of cardiac hypertrophy at the molecular level in consistence with the echocardiographic results (Figure 2 and Figure 3c–e, Table 3). Notably, the losartan-treated RT group showed a trend to a decrease (30%, *p* = 0.139) in β-MHC expression compared to the RT only group, in accordance with the echocardiographic results and supporting the anti-hypertrophic effects of losartan in our RIHD model (Figure 3d and Table 3).

### 2.5. Interstitial Fibrosis Was Reduced in the Losartan-Treated Animals 15 Weeks after RT

To further characterize the anti-remodeling effects of losartan in RIHD, fibrosis was quantified on picrosirius red and fast green-stained sections, and the left ventricular gene expression changes of fibrosis and heart failure markers were measured by qRT-PCR (Figure 1 and Figure 4).

At week 1, there was no significant difference in the collagen content between the groups (Figure 4a,b). Moreover, at week 1, left ventricular expressions of collagen type I alpha 1 (*Col1a1*) and the fibrosis marker connective tissue growth factor (*Ctgf*) showed no significant difference between the control and RT only groups (Figure 4c). Interestingly, losartan significantly reduced the *Ctgf* expression compared to the control group at week 1 (Figure 4c).

At week 3, there were no significant differences in the collagen content and *Col1a1* expressions between the groups, suggesting no severe fibrosis in this stage of RIHD (Figure 4a,b,d). Interestingly, the expression of the pro-fibrotic *Ctgf* was significantly increased in the RT only group and tendentiously elevated (43%, *p* = 0.074) in the losartan-treated RT group compared to the control group (Figure 4d), probably due to initiating wound healing by fibrosis after RT. 

At week 15, the collagen content and *Col1a1* and *Ctgf*, expressions were significantly increased in the RT only group compared to that in the control group, supporting the development of left ventricular fibrosis (Figure 4a–d). There was no significant difference in the collagen content between the control and the losartan-treated RT groups, indicating the anti-fibrotic effects of losartan in RIHD at week 15 (Figure 4a,b). Indeed, losartan significantly reduced the overexpression Col1a1 and Ctgf compared to the RT only group, showing its anti-remodeling effects in our RIHD model at the molecular level (Figure 4c–f).

### 2.6. Losartan Reduced the Chymase Overexpression at Weeks 3 and 15 after RT

Inflammatory processes triggered by the over-activation of RAAS are major contributors to the development of cardiac remodeling and fibrosis in RIHD [8]. Therefore, the effects of losartan on the cardiac expression of the inflammatory cytokines interleukin-1 (*Il1*), interleukin-6 (*Il6*), and tumor necrosis factor-α (*Tnfα*) were measured by qRT-PCR.

At week 1, the expressions of *Il1* and *Il6* were significantly higher in both RT groups, and the expression of *Tnfα* was tendentiously increased (79%, *p* = 0.062) in the RT only group and significantly increased in the RT plus losartan group as compared to the control group, indicating tissue inflammation after RT (Figure 5a–c). At week 3, all measured inflammatory markers, including *Il1*, *Il6*, and *Tnfα*, were significantly overexpressed in the RT only group compared to the control group, pointing out the presence of tissue inflammation at week 3 also (Figure 5a–c). Nevertheless, losartan significantly reduced the overexpression of Il6 at week 3 (Figure 5b). In contrast, the expression of *Il1* and *Tnfα* remained significantly higher in the RT plus losartan group compared to the control group at week 3 (Figure 5a,c). At week 15, there were no significant differences in the *Il1* and *Tnfα* expressions between the groups (Figure 5a–c). However, it should be mentioned that losartan reduced the Tnfα expression in a non-significant manner (45%, *p* = 0.348) compared to the control group. Only *Il6* was significantly overexpressed in the RT groups irrespective of losartan treatment compared to the control group (Figure 5b). Mast cell chymase is an alternative activator of tissue AngII in the heart under inflammatory conditions [23,24]. The collagenase matrix metalloprotease 2 (*Mmp2*) and the mast cell chymase (*Cma*) could be activated by each other in the heart under inflammatory conditions [25]. Indeed, due to the cardiac inflammation in or RIHD model, *Mmp2* was overexpressed in the RT groups irrespective of losartan treatment compared to the time-matched control groups at weeks 1, 3, and 15, respectively (Figure 5d). In contrast, *Cma* was significantly repressed in both RT groups irrespective of losartan treatment compared to the control group at week 1 (Figure 5e). At weeks 3 and 15, the significant overexpression of *Cma* due to RT was markedly reduced in the RT plus losartan group (Figure 5e).

### 2.7. Losartan Alleviated the Cardiac Fibrosis via Inhibiting the TGF-β-Mediated SMAD-Dependent Pathway in Our RIHD Model at Weeks 3 and 15

To further characterize the anti-remodeling effects of the AT1 receptor blocker losartan in RIHD, the expression of angiotensinogen (*Agt*) and transforming growth factor-β (*Tgfb*) was measured by qRT-PCR. In addition, the expression of the main AngII receptors, including the pro-inflammatory, pro-hypertrophic, and pro-fibrotic AT1 receptor, and the anti-inflammatory, anti-hypertrophic and anti-fibrotic type 2 (AT2) receptor were studied at the protein level [14,15] (Figure 6a–f). AT1 receptor was described to activate TGF-β, which can induce fibrosis via the canonical SMAD-dependent and the non-canonical SMAD-independent signaling pathways [14,15,26].

At week 1, there was no significant difference in *Agt* expression between the control and RT only groups. However, in the RT plus losartan group, significantly increased *Agt* expression was found as compared to the control group at week 1 (Figure 6a). No significant differences were detected in the expression of AT1 and AT2 receptors, TGF-β receptor type II (TGF-βRII) and SMAD2/3 levels between the groups at week 1, in consistence with the histology results on either collagen content or cardiomyocyte cross-sectional areas at week 1 (Figure 6b,c,e,f). Interestingly, *Tgfb* was significantly repressed in both RT groups irrespective of losartan compared to the control group at week 1 (Figure 4 and Figure 6d).

At week 3, the *Agt* expression was tendentiously increased (40%, *p* = 0.118) in the RT only group compared to the control group, which may suggest a mild activation of tissue RAAS (Figure 6a). In both RT groups, the AT1 receptor protein levels showed a trend to decrease (19%, *p* = 0.054 and 16%, *p* = 0.090, respectively) compared to the control group (Figure 6b,c). Moreover, in both RT groups, the AT2 receptor protein expressions were significantly increased as compared to the control group (Figure 6c). There was no significant difference in the cardiac *Tgfb* expression between the groups (Figure 6d). In contrast, TGF-βRII and SMAD2/3 protein levels were significantly increased in the RT only group compared to the control group (Figure 6e,f). Losartan significantly reduced the SMAD2/3 levels but not the TGF-βRII level compared to the RT only group at week 3 (Figure 6e,f).

At week 15, there was no significant difference in *Agt* expression between the groups (Figure 6a). The AT1 receptor protein level was significantly decreased in the RT only group compared to the control group at week 15, supposedly, in response to the over-activation of AngII (Figure 6b). There was no significant difference in the AT1 receptor protein levels between the control and losartan-treated RT groups at week 15 (Figure 6b). There were no significant differences between the AT2 protein levels between the groups (Figure 6c). Interestingly, the significant overexpression of *Tgfb* was markedly reduced by losartan at week 15 (Figure 6d). Accordingly, SMAD2/3 protein levels were significantly increased in the RT only group compared to the control group at week 15, suggesting that the canonical SMAD-dependent signaling pathway might play a crucial role in the development of fibrosis in RIHD (Figure 6f). Interestingly, there were no significant differences between the AT2 receptor and TGF-βRII levels between the groups at week 15 (Figure 6c,e). In accordance with the echocardiography and histology results, there was no significant difference in the SMAD2/3 protein level between the losartan-treated RT and the control groups at week 15.

### 2.8. ERK1,2- and AKT-Mediated Pathways Might Be Involved in Compensatory Hypertrophy after RT at Week 15

The non-canonical SMAD-independent signaling pathways include ERK1,2, AKT, and STAT3-mediated fibrotic and hypertrophic pathways beyond other mediators [27]. The levels of total (t)ERK1,2, AKT, and STAT3 as well as their phosphorylated (p) forms are presented in the Appendix A.

At week 1, there was no significant difference in the expression of the pERK1,2/tERK1,2, pAKT/tAKT, and pSTAT3/STAT3 ratios between the groups (Figure 7a–d). At week 3, there were no significant differences in the pERK1/tERK1 and pERK2/tERK2 ratios between the RT only and control groups (Figure 7a,b). In contrast, after losartan treatment at week 3, the pERK1/tERK1 ratio was significantly increased, and a trend to an increase (60%, *p* = 0.072) was seen in the case of the pERK2/tERK2 ratio (Figure 7a,b). The higher pERK1 and pERK2 protein levels led to the increased pERK1/tERK1 and pERK2/tERK2 ratios in the losartan-treated RT group compared to the control group at week 3 (Appendix A). At week 15, pERK1/tERK1 and pERK2/tERK2 ratios were significantly increased in the RT groups irrespective of losartan treatment as compared to the control group (Figure 7a,b). Moreover, in response to losartan, the pERK1/tERK1 ratio was significantly elevated, and the pERK2/tERK2 ratio showed a trend to an increase (61%, *p* = 0.063) as compared to the RT only group (Figure 7a,b). The pERK1 and pERK2 protein levels showed a similar pattern to the pERK/tERK ratios (Appendix A). At weeks 1 and 3, there were no significant differences in the pAKT/tAKT ratios, pAKT, or tAKT levels between the groups (Figure 7c and Appendix A). At week 15, the pAKT/tAKT ratio was significantly higher in both RT groups irrespective of losartan treatment due to the higher pAKT levels compared to that in the control group (Figure 7c and Appendix A).

STAT3 could be activated not only via TGF-β on the non-canonical non-SMAD-dependent fibrotic pathway but also by AngII [28]. There were no significant differences in the pSTAT3/tSTAT3 ratios between the groups at weeks 1 and 15 (Figure 6d). However, in the RT plus losartan group at week 3, the pSTAT3/tSTAT3 ratio was significantly reduced as compared to the control group at week 3 (Figure 7d and Appendix A). Notably, there was no significant difference in pSTAT3 levels between the losartan-treated RT and control groups at week 15, supporting the anti-remodeling effects of losartan in our RIHD model (Appendix A).

## 3. Discussion

Here we report that the selective AT1 receptor blocker losartan alleviates the radiation-induced left ventricular hypertrophy and fibrosis. The antiremodeling effects of losartan seem to be associated with the repression of chymase and several elements of the canonical TGF-β/SMAD2/3 fibrotic pathway in our RIHD model. Moreover, the non-canonical SMAD-independent pathways, including the AKT and ERK1,2-mediated mechanisms, seem to be involved in the maintenance of compensatory hypertrophy in the late phase of RIHD.

RT-induced diffuse myocardial injury, leading to the clinical entity of cardiomyopathy and HF, is a progressive disease ending in cardiac fibrosis, which might develop over several years post-RT depending primarily on RT dose [8,12]. RT-induced cardiomyopathy and HF cover a spectrum of functional abnormalities. Among them, a typical initial phase is HFpEF characterized by DD and compensatory LVH [29,30,31]. Later on, due to the progression of interstitial fibrosis and loss of cardiomyocytes, the clinical entity of HFrEF develops [32,33]. Regarding the underlying pathomechanisms, in humans, radiation injury affects two important systems: one is blood circulation, and the other is the system of cardiomyocytes. Radiation exposure of the coronary arteries and microvasculature results in endothelial cell damage and disturbed circulation. Hypoxia and the injury of the endothelial cells aggravate chronic/progressive inflammation and fibrosis which are essential components of RIHD. In fact, in humans, this component is of utmost significance because RIHD manifests mostly as a potentially fatal coronary event. Concerning radiogenic cardiomyopathy, a fractionated RT dose of 40 Gy to the heart has been found to induce diffuse myocardial injury in patients that may be revealed early by serial follow-up tests [8,32]. Diffuse myocardial injury presents more commonly in patients who received higher RT doses (>60 Gy) and/or anthracycline chemotherapy [33,34]. In most RIHD animal models, exclusively cardiac fibrosis could be studied due to the difficulties in developing radiation-induced arteriosclerosis in rodents [11,12,35,36]. This is why most animal studies, including ours, focus on the effects of radiation-induced diffuse myocardial injury and consequential interstitial inflammation and fibrosis only. Previously, we developed a rat model of high dose radiation-induced heart damage using a single shot of selective heart irradiation of 50 Gy to investigate diffuse myocardial injury [37,38]. In the present study, we used this rat model of RIHD to investigate the effects of losartan.

The presented echocardiography and histology findings are consistent with the literature data [30,31,39] and the previous results in our rat model of RIHD [37,38]. We detected DD in the irradiated animals at every pre-specified follow-up time point in our rat model of RIHD. In fact, the molecular markers (of the expression of β-MHC and α-MHC in the LV and their ratios), echocardiography findings, and histological signs of LVH became more severe over time. Interestingly, the weights of the left ventricles were significantly lower in the RT groups as compared to that in the control group throughout the experiment, probably due to the cell death and developmental retardation caused by the RT. Despite the smaller left ventricular weights, the surviving cardiomyocytes might develop compensatory LVH to maintain the ejection fraction and sufficient oxygen supply to the whole body. Compensatory increase in the red blood cell counts at week 1 might be related to the hypoxia caused by radiation-induced heart and lung damage. Indeed, at week 1, *Il1* was significantly overexpressed in the left ventricles, pointing to the development of tissue inflammation in RIHD. The systemic inflammatory marker white blood cell count was significantly increased at week 3 in the irradiated animals. Accordingly, at week 3, the left ventricular expression of all investigated inflammatory markers, including *Il1*, *Il6*, and *Tnf**α*, were significantly higher post-RT, indicating a more severe tissue inflammation at this phase. At week 15, *Il6* expression (which is a well-known mediator of myocardial fibrosis leading to concentric hypertrophy and secondary DD [40]) remained high. In the fibrotic phase with severe concentric LVH at week 15, hemoglobin and hematocrit levels significantly increased, probably, in response to the chronic hypoxia caused by radiogenic heart failure and lung fibrosis.

The key molecular mechanism of radiation-induced cardiac fibrosis is thought to be the TGF-β/SMAD2/3-mediated fibrotic pathway [12,21,41,42]. Indeed, molecular signs (i.e., overexpression of *Ctgf,* TGF-βRII, and SMAD2/3) preceded the histologic signs of fibrosis 3 weeks after the irradiation, showing the initiation of fibrotic remodeling in the heart in our RIHD model. At week 15, left ventricular *Ctgf* and SMAD2/3 expressions remained high with the overexpression of *Tgfb* and *Col1a1.* At week 15, the significantly increased interstitial collagen content of the left ventricle revealed the presence of fibrosis in RIHD. These molecular, morphological, and functional changes were similar to those reported in other studies [30,31]. We also investigated selected molecules, including AKT, ERK1,2, and STAT3 and their phosphorylated forms, in the non-canonical non-SMAD-dependent fibrotic pathway. Significantly increased pERK1,2/tERK1,2 and pAKT/tAKT ratios were detected after the RT at week 15. Since the ERK-mediated pathways are also considered survival pathways regulating cell death and hypertrophy [43], the increased pERK1,2/tERK1,2 ratios in the RT groups could be explained as compensatory defense mechanisms against cell death. AKT was also described as a molecule mediating cardiomyocyte survival and hypertrophy [27], and this, in fact, might explain the higher pAKT/tAKT ratios in the RT groups at week 15. In summary, the canonical SMAD-dependent TGF-β pathway seems to be responsible for the cardiac fibrosis in our RIHD model. The non-canonical SMAD-independent pathways, including the AKT and ERK1,2-mediated mechanisms, seem to be involved in cell survival and compensatory hypertrophy in RIHD.

It has been reported that irradiation could induce AngII overexpression in the rat heart [16] and lung [17] in a dose-dependent manner [18]. AngII can be cleaved from AngI by several proteases, of which angiotensin-converting enzyme (ACE), and alternatively, under inflammatory conditions, mast cell chymase are the primary converters [23,44,45]. Chymase overexpression seems to be particularly important in the cardiac generation of AngII in RIHD [23,46,47]. Interestingly, the cardiac RAAS interacts with many other systems in the heart, including the endothelin system and cardiac sympathetic nervous systems. The locally generated AngII appears to contribute to LVH and fibrosis via different mechanisms [23]. AngII has been reported to activate cardiac NADPH-oxidase via AT1 receptor, and subsequently, the over-production of reactive oxygen and nitrogen species leads to increased nitro-oxidative stress [48,49]. This could trigger the production of pro-inflammatory mediators, such as IL1, IL6, TNF-α, and TGF-β, contributing to cardiac remodeling and heart failure [48,49]. The AT1 receptor is mainly involved in pro-inflammatory, pro-hypertrophic, and pro-fibrotic mechanisms, whereas the AT2 receptor is associated with counter-regulatory anti-inflammatory, anti-hypertrophic and anti-fibrotic pathways in the cardiovascular system [14,15]. Therefore, selective AT1 blockade by losartan seems to be a rational therapeutic option to ameliorate cardiac remodeling by reducing nitro-oxidative stress and inflammatory mechanisms in RIHD. However, to our best knowledge, there are no experimental data on the anti-remodeling effects of ARBs in RIHD. Interestingly, two clinical trials investigated the effects of ARBs after RT; however, their results were controversial. The PRADA clinical trial demonstrated that administration of the ARB candesartan improved the reduced ejection fraction in early breast cancer patients treated with anthracycline-containing regimens with or without trastuzumab and RT [19]. In contrast, a retrospective clinical study found that ACE inhibitors or ARBs and higher RT doses were related to lower overall survival after the diagnosis of cancer in those patients who were treated with thoracic RT and later underwent percutaneous coronary interventions [20].

In the present study, at week 1, losartan-treatment did not influence the early molecular (i.e., left ventricular expressional changes of β-MHC and α-MHC and their ratios) and echocardiographic signs of LVH as assessed by M-mode echocardiography, probably due to the acute compensatory and surviving mechanisms in response to tissue damage caused by RT. At week 3, in the losartan-treated animals, various experimental parameters improved as compared to that in the RT only group, such as several echocardiographic signs of LVH and consequential DD together with the overexpression of β-MHC and SMAD2/3 in the left ventricle indicating a possible anti-hypertrophic and anti-fibrotic effect of losartan in the early chronic phase of RIHD. In contrast, white blood cell counts remained elevated irrespective of losartan treatment in RIHD at week 3, supposing the presence of systemic inflammation. At week 15, losartan treatment did not reduce the molecular signs of LVH (the changes in the expression of β-MHC and α-MHC and their ratios), suggesting that the hypertrophic process is also active in this late chronic phase of RIHD as a compensatory mechanism to cardiomyocyte loss after RT and secondary hypoxia. However, according to the echocardiography and histology findings at week 15, losartan reduced the severity of LVH. Losartan reduced the interstitial collagen content by reducing the *Ctgf, Tgfb,* and *Col1a1* overexpression and SMAD2/3 levels in RIHD. Similar to our results, losartan reduced cardiac fibrosis in heart failure by reducing the CTGF and SMAD2/3 expression in other studies [50,51]. We also investigated molecules in the non-canonical SMAD-independent fibrotic pathway and found that pERK1,2/tERK1,2 ratios were significantly increased in the RT groups and were further increased by losartan at week 15. In contrast, several studies found that losartan reduced the pERK1,2/tERK1,2 ratio in heart failure [52,53]. Our result could be explained by the fact that ERK1,2 phosphorylation can be increased by MMP2 [54]. Indeed, losartan failed to significantly reduce the overexpression of *Mmp2* in our RIHD model.

Interestingly, at weeks 3 and 15, angiotensinogen expression was not significantly different between the groups, but the AT1 receptor levels were decreased, probably, due to increased AngII levels after RT. This finding is in accordance with human data showing that AT1 receptors are selectively downregulated in heart failure depending on the severity of ventricular dysfunction [55]. At week 15, the repression of the AT1 receptor was weakened by losartan, supposedly via blocking the cleavage of AngII from AngI. Indeed, the left ventricular expression of the mast cell chymase was higher both at week 3 and 15 in RIHD, likely mediated by cardiac inflammation and *Mmp2* overexpression [56]. Chymase overexpression was reduced by losartan at weeks 3 and 15, possibly explained by its mild anti-inflammatory effects in RIHD [57] (i.e., repression of *Il6* at week 3, and decreased expression of *Tnfα* at week 15). Indeed, it has been described that losartan was more effective than the ACE-inhibitor captopril in controlling ongoing vascular inflammation if AngII-dependent components of atherogenesis were present in mice [45]. However, DD was not improved by losartan in our RIHD model at week 15, which may have been due to the presence of local inflammation (i.e., *Il6* overexpression) and more severe LVH and fibrosis at this late chronic stage of RIHD.

Our study is not without limitations, similar to other experimental works. We intended to test losartan for controlling heart failure related to RIHD. Nevertheless, significant differences exist between the pathomechanisms in the model used vs. that in patients. (i) While, in humans, radiogenic coronary artery damage is an essential component of the radiation damage, in most murine models, no similar alterations evolve unless specific genetically modified animal strains are used [10,11,12,41,46]. In our experiments, RT-induced diffuse myocardial injury and consequential interstitial inflammation and fibrosis were the basis of the heart failure studied [37]. (ii) In order to obtain the measurable end-points of heart failure and interstitial fibrosis in the myocardium [8,12,33,41] within a reasonable timeframe, we used a very high biological dose with drastic consequences to which a similar one is never applied in conventional radiotherapy. Nevertheless, in modern radiotherapy practice, the use of very large doses (stereotactic radiosurgery delivered with precision selectivity) is widely utilized [58]. (iii) In patients, the effects of various risk factors (such as hyperlipidemia, diabetes, hypertension, aging, sex hormonal differences, individual radiosensitivity, etc.) may modulate radiation-induced changes, which are less pronounced in most animal models. (iv) Finally, while in the animal model, a relatively short follow-up time was used post-radiotherapy, in patients, about 10 years elapse until the recognition of RIHD, although the presence of manifest heart disease or other risk factors may advance its diagnosis [6,7,11,35]. Moreover, we used only one dose of losartan which is comparable to the human therapeutic doses and widely used in heart failure models of other etiologies. The demonstration of molecular mechanisms of how losartan interacts with tissue remodeling was out of the scope of our present descriptive study. We found here anti-remodeling effects of losartan and hypothesized that losartan could have anti-fibrotic effects by ameliorating the TGF-β, /SMAD-mediated pathway and reducing the overexpression of chymase. Moreover, the non-canonical SMAD-independent pathways, including the AKT and ERK1,2-mediated mechanisms, seem to be involved in maintaining compensatory hypertrophy in our chronic RIHD model. Inhibition of the TGF-β /SMAD pathway or the non-SMAD-dependent pathways in further experiments is needed to determine their exact role in losartan-mediated anti-remodeling effects in RIHD.

## 4. Materials and Methods

### 4.1. Ethics Approval

The study was conducted according to the guidelines of the Declaration of Helsinki, and approved by the University of Szeged and the regional Animal Research Ethics Committee of Csongrád County (Csongrád county, Hungary; project license: XV./800/2019, date of approval: 30 May 2019). All institutional and national guidelines for the care and use of laboratory animals were followed.

### 4.2. Animals

A total of 63 male Sprague-Dawley rats (220–300 g, 6–7 weeks old) were used in three separate experiments (*n* = 6–9 in each group). After 1 week of acclimatization in a temperature-controlled room (22 ± 2 °C; relative humidity 55 ± 10%), the animals were randomly assigned to the control or the two RT groups and into 3 subgroups, each with 1, 3, or 15 weeks follow-up times, respectively. A total of 21 animals served as controls, and a total of 42 animals received a single dose of 50 Gy delivered to the whole heart to induce RIHD as described previously [37,38]. The animals were housed in pairs in individually ventilated cages (Sealsafe IVC system, Buguggiate, Italy) in a temperature-controlled room with a 12 h:12 h light/dark cycle. Standard rat chow and tap water were supplied ad libitum.

### 4.3. Experimental Setup

Rats were divided into three groups (*n* = 6–9 in each group, Figure 1) and treated via oral gavage daily for 1, 3, and 15 weeks, respectively, as follows: (i) control group: treated with tap water (per os 2 mL/kg/day, *n* = 7), (ii) RT only group: treated with tap water (per os 2 mL/kg/day, *n* = 6–7), and (iii) RT plus losartan group: treated with losartan (per os 10 mg/kg/day dissolved in tap water in 2 mL/kg end volume, Arbartan 50 mg film-coated tablets, Teva Pharmaceutical Industries Ltd., Debrecen, Hungary, *n* = 7–9). Cardiac morphology and function were assessed by transthoracic echocardiography at the end-point of each experiment (Figure 1). At the end of the different follow-up times, rats were anesthetized with sodium pentobarbital (Euthasol, ip. 40 mg/kg Produlab Pharma b.v., Raamsdonksveer, The Netherlands). Then blood was collected from the abdominal aorta to measure routine laboratory parameters (Figure 1). After the blood sampling, hearts, lungs, and tibias were isolated, and the blood was washed out from the heart in calcium-free Krebs-Henseleit solution. Then left and right ventricles were separated, and left ventricular samples were prepared for histology and biochemical measurements. The development of LVH and fibrosis in the irradiated groups were verified by the measurement of cardiomyocyte cross-sectional areas on hematoxylin-eosin (HE)-stained slides and picrosirius red/fast green-stained (PSFG) slides (Figure 1). Total RNA was isolated from the left ventricles, and the expression of hypertrophy and fibrosis (i.e., *Myh6*, *Myh7*, *Ctgf*, *Tgfb*, *Col1a1*, and *Mmp2*), RAAS-associated (i.e., *Cma* and *Agt*), inflammatory (i.e., *Il1, Il6*, and *Tnf*) markers were measured at the transcript level by qRT-PCR in every time point (Figure 1). Moreover, left ventricular protein levels of AT1R, AT2R, TGF-βRII, SMAD2/3, STAT3, pSTAT3, AKT, pAKT, ERK1, ERK2, pERK1, pERK2 were measured by using Western blot technique at weeks 1, 3, and 15.

### 4.4. Heart Irradiation

Heart irradiation with a single dose of 50 Gy in the groups of RT and RT plus losartan was carried out as described previously [37,38]. Before the irradiation, rats were anesthetized with sodium pentobarbital (Euthasol, *ip*. 40 mg/kg, Produlab Pharma b.v., Raamsdonksveer, The Netherlands), then fixed in the supine position to a flat surface couch. Briefly, the planning of the irradiation was based on a 3D model, and the dose was delivered to the geometric centre of the heart. For better coverage of the heart and lung protection, a 6 MeV electron radiation was given with a circle-shaped aperture with a 2 cm diameter. The radiation dose was delivered with a Primus linear accelerator (Siemens Healthcare GmbH, Erlangen, Germany) at a dose intensity of 5 Gy/min if the appropriate position of the animal was proven using a built-in electronic portal imaging device.

### 4.5. Transthoracic Echocardiography

Cardiac morphology and function were assessed by transthoracic echocardiography as described previously [59,60], at weeks 1, 3, and 15 to monitor the development of RIHD. Rats were anesthetized with 2% isoflurane (Forane, Aesica, Queenborough Limited, Queenborough, UK). Two-dimensional, M-mode, Doppler, tissue Doppler, and four chamber-view images were performed by the criteria of the American Society of Echocardiography with a Vivid IQ ultrasound system (General Electric Medical Systems, New York, NY, USA) using a phased array 5.0–11 MHz transducer (General Electric 12S-RS probe, General Electric Medical Systems, New York, NY, USA). Data of three consecutive heart cycles were analyzed (EchoPac Dimension v201, General Electric Medical Systems, USA; https://www.gehealthcare.com/products/ultrasound/vivid/echopac, accessed on 29 November 2021) by an experienced investigator in a blinded manner. The mean values of three measurements were calculated and used for statistical evaluation. Systolic and diastolic wall thicknesses were obtained from parasternal short-axis view at the level of the papillary muscles (in cases of anterior and inferior walls) and long-axis view at the level of the mitral valve (in cases of septal and posterior walls). The left ventricular diameters were measured by means of M-mode echocardiography from long-axis and short-axis views between the endocardial borders. The fractional shortening was calculated on M-mode images in the long-axis view. Diastolic function was assessed using pulse-wave Doppler across the mitral valve and tissue Doppler on the septal mitral annulus from the apical four-chamber view. Early (E) flow and septal mitral annulus velocity (e’) indicate diastolic function. Ejection fraction was calculated on four-chamber view images using the modified Simpson method (i.e., biplane method of disks) requiring area tracings of LV cavity in end-systole and end-diastole.

### 4.6. Blood Parameters

Blood was collected from the abdominal aorta at weeks 1, 3, and 15. Total blood count and hematocrit were measured from whole blood by a hematology analyzer (XE-2100, Sysmex Corporation, Kobe, Japan) [60] to characterize the severity of systemic inflammation and the compensatory increase in red blood cell synthesis associated with lung and heart damage due to RT.

### 4.7. Tissue Harvesting

At weeks 1, 3, and 15, the hearts of the animals of the respective subgroups were isolated under pentobarbital anesthesia (Euthasol, ip. 40 mg/kg, Produlab Pharma b.v., Raamsdonksveer, The Netherlands), and the blood was washed out in calcium-free Krebs-Henseleit solution. Then the hearts were weighed, left and right ventricles were separated and weighed, and the cross-section of the left ventricle at the ring of the papillae was cut and fixed in 4% buffered formalin for histological analysis. Other parts of the left ventricles were freshly frozen in liquid nitrogen and stored at −80 °C until further biochemical measurements. Body weight, tibia length, and weights of the lungs were also measured.

### 4.8. Hematoxylin-Eosin and Picrosirius Red and Fast Green Stainings

Five μm paraffin-embedded transverse cut sections of the formalin-fixed subvalvular areas of the left ventricles were stained with hematoxylin-eosin (HE) or picrosirius red and fast green (PSFG) as described previously [38,61]. Histological slides were scanned with a Pannoramic Midi II scanner (3D-Histech, Budapest, Hungary). On the digital HE images, cardiomyocyte cross-sectional areas were measured to verify the development of LVH at the cellular level. For the evaluation, the Biology Image Analysis Software (BIAS) was used [60]. BIAS (internal built, dated December 2019; https://single-cell-technologies.com/bias/, accessed on 29 November 2021) is developed by Single-Cell Technologies Ltd., Szeged, Hungary, and the first publicly available version is expected to be released in late 2021. Image pre-processing was followed by deep learning-based cytoplasm segmentation. User-selected objects were forwarded to the feature extraction module, configurable to extract properties from the selected cell components. Here, transverse transnuclear cardiomyocyte perimeters were measured in 100 (consecutive) cardiomyocytes selected on the basis of longitudinal orientation and mononucleation from a single cut-surface (digitalized histological slide) of the left ventricular tissue blocks. Cardiac fibrosis was assessed on PSFG slides with an in-house developed program [38,61]. Briefly, this program determines the proportion of red pixels of heart sections using two simple color filters. For each red–green–blue (RGB) pixel, the program calculates the color of the pixel in hue–saturation–luminance (HSL) color space. The first filter is used for detecting red portions of the image. The second filter excludes any white (empty) or light grey (residual dirt on the slide) pixels from further processing using a simple RGB threshold. In this way, the program groups each pixel into one of two sets: pixels considered red and pixels considered green but neither white, nor grey. Red pixels in the first set represent collagen content and fibrosis. Green pixels in the second set correspond to cardiac muscle. The mean values of 10 representative images were calculated and used for statistical evaluation in the case of each left ventricular slide. Medium-size vessels and their perivascular connective tissue sheet, the subepicardial and subendocardial areas were avoided as much as possible. Representative HE- and PSFG-stained slides were captured in Panoramic Viewer 1.15.4 (3D-Histech, Budapest, Hungary; https://old.3dhistech.com/pannoramic_viewer, accessed on 29 November 2021).

### 4.9. mRNA Expression Profiling by qRT-PCR

Quantitative RT-PCR was performed with gene-specific primers to monitor mRNA expression as described previously [59,60]. RNA was isolated using Qiagen RNeasy Fibrous Tissue Mini Kit (Qiagen, Hilden, Germany) from heart tissue. Then 100 μg of total RNA was reverse transcribed using iScript™ cDNA Synthesis Kit (BioRad Laboratories Inc., Hercules, CA, USA). Specific primers (*Agt*: angiotensinogen, #qRnoCED0051666; *Cma1*: chymase, #qRnoCED0005462; *Col1a1*: collagen type 1 alpha 1 chain, #qRnoCED0007857; *Ctgf*: connective tissue growth factor, #qRnoCED0001593; *Il1*: interleukin-1, #qRnoCID0002056; *Il6*: interleukin-6, #qRnoCID0053166; *Mmp2:* matrix metalloproteinase 2, #qRnoCID0002887; *Myh6*: α-myosin heavy chain, #qRnoCID0001766; *Myh7*: β-myosin heavy chain, #qRnoCED0001215; *Tgfb*: transforming growth factor-β, #qRnoCID0009191, *Tnf-α*: tumor necrosis factor-α, #qRnoCED0009117) and SsoAdvanced™ Universal SYBR^®^ Green Supermix (BioRad Laboratories Inc., Hercules, CA, USA) were used according to the manufacturer’s instructions. Ribosomal protein lateral stalk subunit P2 (*RpIp2*, forward primer sequence: *agcgccaaagacatcaagaa* and reverse primer sequence: *tcagctcactgatgaccttgtt*) was used as a housekeeping control gene for normalization.

### 4.10. Western Blot

To investigate gene expression changes at the protein level, a standard Western blot technique was used as described previously [59,60]. AT1R (41 kDa), AT2R (41 kDa), SMAD2/3 (52 and 60 kDa), TGF-βRII (85 kDa), STAT3 (79 and 86 kDa), pSTAT3 (79 and 86 kDa), AKT (60 kDa), pAKT (60 kDa), ERK1/2 (42 and 44 kDa), and pERK1/2 (42 and 44 kDa) with GAPDH (37 kDa) loading background were assessed at weeks 1, 3, and 15. Left ventricular samples (*n* =6–7 in each group, total *n* = 21 at week 1, total *n* = 20 at week 3, and total *n* = 20 at week 15) were homogenized with an ultrasonicator (UP100H, Hielscher, Germany) in Radio-Immunoprecipitation Assay (RIPA) buffer (50 mM Tris-HCl (pH 8.0), 150 mM NaCl, 0.5% sodium deoxycholate, 5 mM ethylenediamine tetra-acetic acid (EDTA), 0.1% sodium dodecyl sulfate, 1% NP-40; Cell Signaling Technology Inc., Danvers, MA, USA) supplemented with phenylmethanesulfonyl fluoride (PMSF; Sigma-Aldrich, St. Louis, MO, USA) and sodium fluoride (NaF; Sigma-Aldrich, Saint Louis, MO, USA). The crude homogenates were centrifuged at 15,000× *g* for 30 min at 4 °C. After quantifying the supernatants’ protein concentrations using the BCA Protein Assay Kit (Pierce Thermo Fisher Scientific Inc., Waltham, MA, USA), 25 μg of reduced and denaturized protein was loaded. Then sodium dodecyl-sulfate polyacrylamide gel electrophoresis (SDS-PAGE, 50 V, 4 h) was performed (10% gel in case of AT1R, AT2R, SMAD2/3, TGF-βRII, STAT3, phospho-STAT3, AKT, phospho-AKT, ERK1/2, and phospho-ERK1/2) followed by the transfer of proteins onto a nitrocellulose membrane (10% methanol in case of AT1, AT2, SMAD2/3, TGF-βRII and 20% methanol in case of STAT, phospho-STAT, AKT, phospho-AKT, ERK1/2, phospho-ERK1/2, 35 V, 2 h). The efficacy of transfer was checked using Ponceau staining. The membranes were cut horizontally into parts corresponding to the molecular weights of AT1R, AT2R, SMAD2/3, TGF-βRII, STAT3, pSTAT3, AKT, pAKT, ERK1/2, pERK1/2, and GAPDH. Membranes were blocked for 1 h in 5% (*w*/*v*) bovine serum albumin (BSA, Sigma-Aldrich, Saint Louis, MO, USA) and were incubated with primary antibodies in the concentrations of 1:1000 against AT1R (#ab124734, Abcam PLC, Cambridge, UK), AT2R (#ab92445, Abcam PLC, Cambridge, UK), SMAD2/3 (#8685T, Cell Signaling Technology Inc., Danvers, MA, USA), TGF-β receptor II (#79424T, Cell Signaling Technology Inc., Danvers, MA, USA), STAT3 (#9139, Cell Signaling Technology Inc., Danvers, MA, USA), pSTAT3 (#9145, Cell Signaling Technology Inc., Danvers, MA, USA), AKT (#2920, Cell Signaling Technology Inc., Danvers, MA, USA), pAKT (#4060, Cell Signaling Technology Inc., Danvers, MA, USA), ERK1/2 (#4696, Cell Signaling Technology Inc., Danvers, MA, USA), pERK1/2 (#9101S, Cell Signaling Technology Inc., Danvers, MA, USA), or 1:5000 against GAPDH (#2118, Cell Signaling Technology Inc., Danvers, MA, USA) overnight at 4 °C in 5% BSA. Then the membranes were incubated with IRDye^®^ 800CW Goat Anti-Rabbit and/or IRDye^®^ 680RD Goat Anti-Mouse secondary antibody (LI-COR Biosciences, Lincoln, NE, USA, in the concentrations of 1:5000) for 1 h at room temperature in 5% BSA antibodies to detect proteins with similar molecular weight on the same membrane where it is applicable. Fluorescent signals were detected by Odyssey CLx machine (LI-COR Biosciences, Lincoln, NE, USA), and digital images were analyzed and evaluated by densitometry with Quantity One Software (Bio-Rad Laboratories Inc., Hercules, CA, USA).

### 4.11. Statistical Analysis

Statistical analysis was performed using Sigmaplot 12.0 for Windows (Systat Software Inc., San Jose, CA, USA). All values are presented as mean ± SEM. Specific sample numbers used for measurements are described in the corresponding figure legend. One-Way ANOVA was used to determine the statistical significance between all measured parameters within each time point. A Holm-Sidak test was used as a post-hoc test. *p* < 0.05 was accepted as a statistically significant difference.

## 5. Conclusions

In this study, we evaluated the effects of chronic administration of the selective AT1 receptor blocker losartan on cardiac remodeling, function, and molecular markers of inflammation, LVH, fibrosis, and heart failure in a rat model of RIHD. Our results suggest that the development of RIHD-associated LVH and fibrosis might be prevented or markedly slowed down by losartan if its administration starts early after RT. Losartan seems to ameliorate left ventricular fibrosis. The antiremodeling effects of losartan seem to be associated with the repression of chymase and several elements of the canonical SMAD-dependent TGF-β signaling pathway in our rat model of RIHD. In contrast, the non-canonical SMAD-independent pathways, including the AKT and ERK1,2-mediated mechanisms, seem to be involved in the maintenance of compensatory hypertrophy in our late phase RIHD model. In order to clarify whether losartan and other ARBs may be protective against RIHD in humans, clinical trials enrolling a large number of patients are required for widening the indication of this otherwise routinely used drug of heart failure treatment.

## Figures and Tables

**Figure 1 ijms-22-12963-f001:**
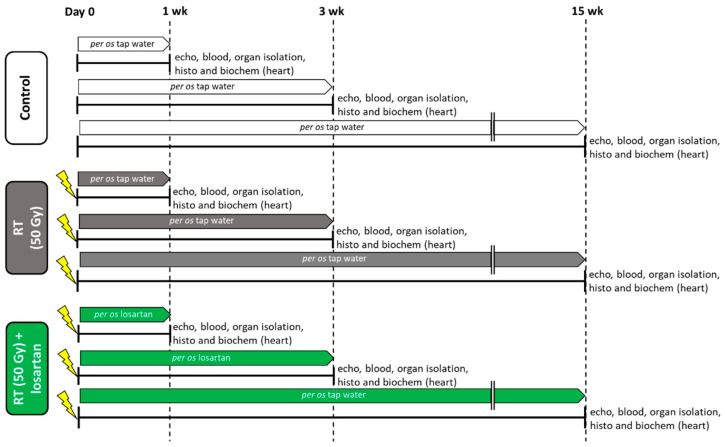
Experimental setup. Rats (*n* = 63) were divided into three groups (*n* = 6–9) and treated via oral gavage daily for 1, 3, or 15 weeks, respectively, as follows: (i) control group treated with tap water, (ii) radiotherapy (RT) only group treated with tap water, and (iii) RT plus losartan group treated with losartan (per os 10 mg/kg/day) dissolved in tap water. Cardiac morphology and function were assessed by transthoracic echocardiography (echo) at the end-points of each experiment under anesthesia. Then, blood was collected from the abdominal aorta to measure routine laboratory parameters, and hearts, lungs, and tibias were isolated. Left and right ventricles were separated, and left ventricular samples were prepared for histology (histo) and biochemical measurements (biochem). The development of LVH and fibrosis in the irradiated groups were investigated by the measurement of cardiomyocyte cross-sectional areas on hematoxylin-eosin-stained slides and picrosirius red/fast green-stained slides. The expression of selected genes related to LVH and fibrosis, heart failure, renin-angiotensin-aldosterone system (RAAS), and inflammation were measured at the transcript level by qRT-PCR. Left ventricular levels of selected proteins related to the RAAS, cardiac hypertrophy, and fibrosis pathways were measured by Western blot.

**Figure 2 ijms-22-12963-f002:**
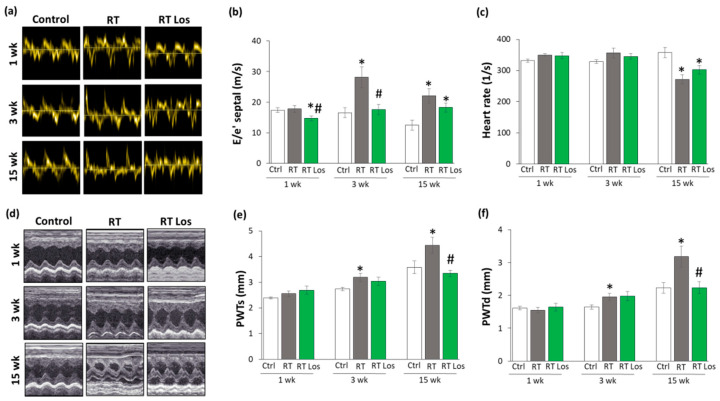
The effects of losartan on the echocardiographic parameters at weeks 1, 3, and 15. (**a**) Representative tissue Doppler images of diastolic septal mitral annulus velocity e’, (**b**) mitral valve early flow velocity (E)/ e’, (**c**) heart rate, (**d**) representative M-mode images of wall thicknesses and left ventricular diameters, (**e**) posterior wall thicknesses in systole (PWTs) and (**f**) diastole (PWTd). Values are presented as mean ± S.E.M., * *p* < 0.05 vs. control group, # *p* < 0.05 vs. RT only group (*n* = 6–7, One-Way ANOVA, Holm-Sidak post hoc test). Ctrl: control group, RT: radiotherapy only group (50 Gy), RT Los: RT plus losartan group. Representative M-mode images were saved from the Echo- Pac Dimension v201 software.

**Figure 3 ijms-22-12963-f003:**
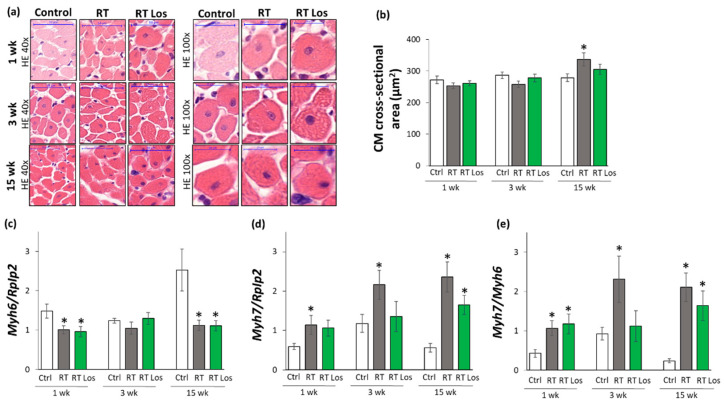
The effects of losartan on left ventricular hypertrophy assessed by histology at weeks 1, 3, and 15. (**a**) Representative hematoxylin-eosin (HE)-stained sections at 40× and 100× magnification, (**b**) cardiomyocyte cross-sectional area, (**c**) alpha-myosin heavy chain (*Myh6*), and (**d**) beta-myosin heavy chain (*Myh7*) expression in the left ventricle normalized to the ribosomal protein lateral stalk subunit P2 (*RpIp2)* gene expression, (**e**) *Myh7/Myh6* ratios. On the digital HE images, cardiomyocyte (CM) cross-sectional areas were measured in 100 selected cardiomyocytes in left ventricular tissue sections cut on equivalent planes. Scale bars represent 50 µm in the 40× magnified images and 20 µm in the 100× magnified images. Values are presented as mean ± S.E.M., * *p* < 0.05 vs. control group, (*n* = 6–7, One-Way ANOVA, Holm-Sidak post hoc test). Ctrl: control group, RT: radiotherapy only group (50 Gy), RT Los: RT plus losartan group. Representative HE-stained slides were captured in the Panoramic Viewer 1.15.4 software.

**Figure 4 ijms-22-12963-f004:**
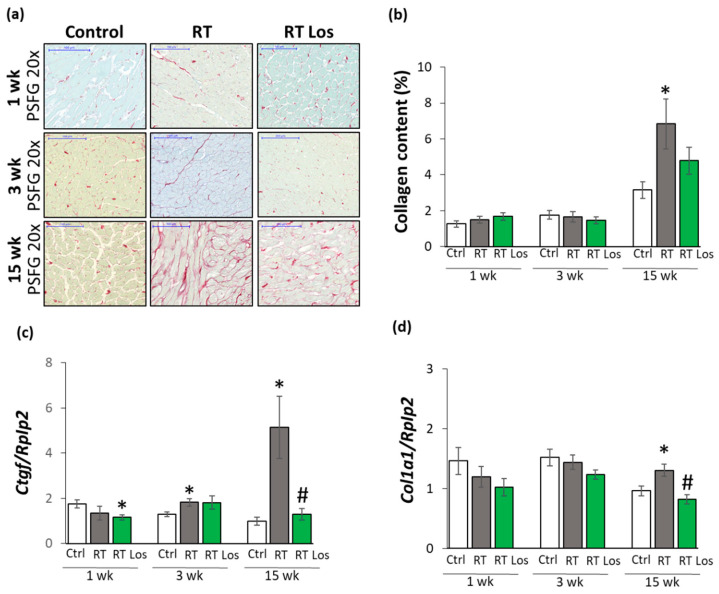
The effects of losartan on left ventricular fibrosis as assessed by histology and qRT-PCR at weeks 1, 3, and 15. (**a**) Representative picrosirius red and fast green (PSFG)-stained sections at 20× magnification, (**b**) left ventricular collagen content, left ventricular expression of (**c**) connective tissue growth factor (*Ctgf*), (**d**) collagen 1a1 (*Col1a1*) normalized to ribosomal protein lateral stalk subunit P2 (*Rplp2*) gene expression. The mean values of the collagen content of 10 representative PSFG-stained images were calculated and used for statistical evaluation in each left ventricular sample. Scale bars represent 100 µm at the 20× magnified images. Values are presented as mean ± S.E.M., * *p* < 0.05 vs. control group, # *p* < 0.05 vs. RT only group (*n* = 6–7, One-Way ANOVA, Holm-Sidak post hoc test). Ctrl: control group, RT: radiotherapy only group (50 Gy), RT Los: RT plus losartan group. Representative PSFG-stained slides were captured in the Panoramic Viewer 1.15.4 software.2.7.

**Figure 5 ijms-22-12963-f005:**
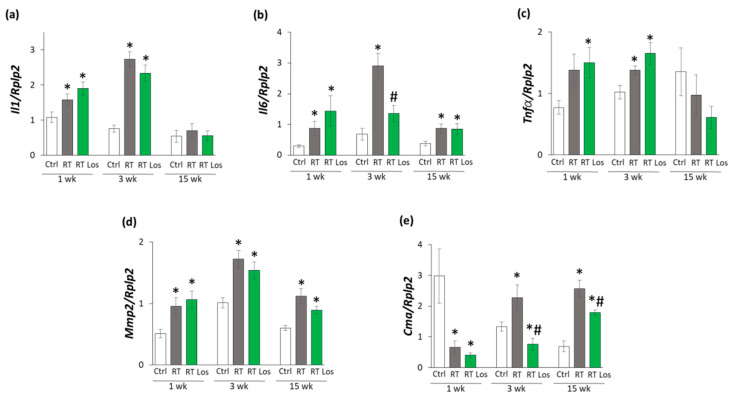
The effects of losartan on inflammatory gene expressions assessed by qRT-PCR at weeks 1, 3, and 15. Expression of (**a**) interleukin-1 (*Il1*), (**b**) interleukin-6 (Il6), (**c**) tumor necrosis factor-alpha (*Tnfα*), (**d**) matrix metalloprotease-2 (*Mmp2*), and (**e**) chymase (*Cma*) normalized to ribosomal protein lateral stalk subunit P2 (*RpIp2*) gene expression were measured in left ventricle samples. Values are presented as mean ± S.E.M., * *p* < 0.05 vs. control group, # *p* < 0.05 vs. RT only group (*n* = 6–7, One-Way ANOVA, Holm-Sidak post hoc test). Ctrl: control group, RT: radiotherapy only group (50 Gy), RT Los: RT plus losartan group.

**Figure 6 ijms-22-12963-f006:**
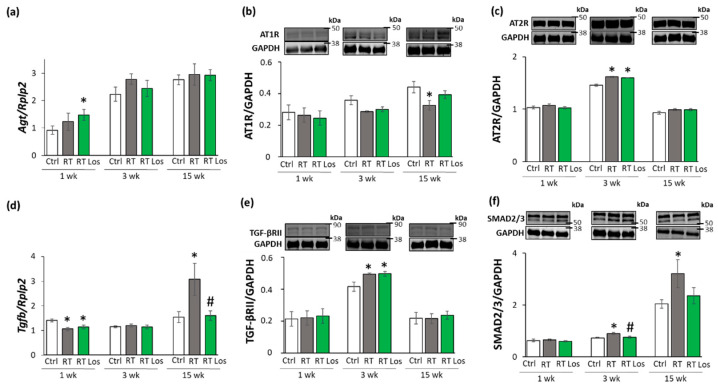
The effects of losartan on the expression of genes and proteins related to the cardiac renin-angiotensin-aldosterone system and canonical SMAD-dependent pathway at weeks 1, 3, and 15. Left ventricular expression of (**a**) angiotensinogen (*Agt*) normalized to ribosomal protein lateral stalk subunit P2 (*RpIp2*) gene expression, left ventricular protein expression and cropped representative images of (**b**) angiotensin II type 1 receptor (AT1R, 41 kDa), and (**c**) angiotensin II type 2 receptor (AT2R, 41 kDa), (**d**) left ventricular expression of transforming growth factor-beta (*Tgfb*) normalized to *RpIp2* gene expression, left ventricular expression and cropped representative images of (**e**) TGF-β receptor II (TGF-βRII, 85 kDa) and (**f**) SMAD2/3 (52 and 60 kDa). Values are presented as mean ± S.E.M., * *p* < 0.05 vs. control group, # *p* < 0.05 vs. RT only group (*n* = 6–7, One-Way ANOVA, Holm-Sidak post hoc test). Ctrl: control group, RT: radiotherapy only group (50 Gy), RT Los: RT plus losartan group. Glyceraldehyde-3-phosphate dehydrogenase (GAPDH, 37 kDa) was used as a loading control in protein expression changes assessed by Western blot. Images were captured with the Odyssey CLx machine and exported with Image Studio 5.2.5 software. Black lines next to the Western blot images represent the position of protein markers with corresponding molecular weights. The uncropped Ponceau-stained membranes and the full-length Western blot images with the protein ladders are presented in Appendix A.

**Figure 7 ijms-22-12963-f007:**
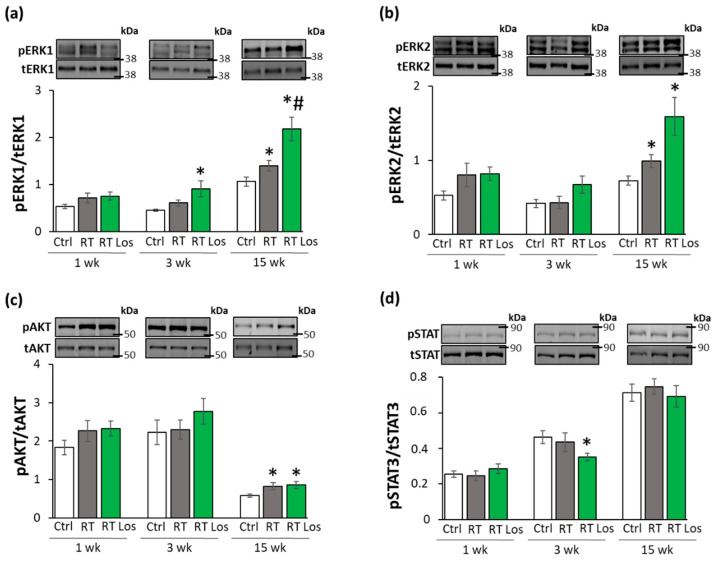
The effects of losartan on the expression of proteins associated with the non-canonical SMAD-independent fibrotic pathway at weeks 1, 3, and 15. Left ventricular expression and cropped representative images of (**a**) phospho-ERK1/total-ERK1 ratio (pERK1/tERK1 42 and 44 kDa), (**b**) phospho-ERK2/total-ERK2 ratio (pERK2/tERK2, 42 and 44 kDa), (**c**) phospho-AKT/total-AKT ratio (pAKT/tAKT, 60 kDa), (**d**) phospho-STAT3/total-STAT3 ratio (pSTAT3/tSTAT3, 79 and 86 kDa). Values are presented as mean ± S.E.M., * *p* < 0.05 vs. control group, # *p* < 0.05 vs. RT only group (*n* = 6–7, One-Way ANOVA, Holm-Sidak post hoc test). Ctrl: control group, RT: radiotherapy only group (50 Gy), RT Los: RT plus losartan group. Images were captured with the Odyssey CLx machine and exported with Image Studio 5.2.5 software. The phospho-protein and total protein levels are represented in Appendix A. Black lines next to the Western blot images represent the position of protein markers with corresponding molecular weights. The uncropped Ponceau-stained membranes and the full-length Western blot images with the protein ladders are presented in Appendix A.

**Table 1 ijms-22-12963-t001:** Post-RT body weights, organ weights, and tibia lengths at weeks 1, 3, and 15.

Parameter (Unit)	Week 1			Week 3			Week 15		
	Control	RT	RT Los	Control	RT	RT Los	Control	RT	RT Los
Body weight at the beginning (g)	233 ± 8	233 ± 5	232 ± 4 *	282 ± 5	281 ± 7	282 ± 6	261 ± 3	263 ± 4	261 ± 5
Body weight at the endpoint (g)	269 ± 7	258 ± 6	251 ± 5 *	382 ± 9.57	338 ± 5.22 *	344 ± 8.8 *	525 ± 14	377 ± 29 *	414 ± 17 *
Tibia length (cm)	3.39 ± 0.06	3.36 ± 0.03	3.33 ± 0.04	3.83 ± 0.07	3.73 ± 0.05	3.86 ± 0.09	4.5 ± 0.04	4.25 ± 0.08 *	4.28 ± 0.04 *
Heart weight (g)	0.91 ± 0.02	0.87 ± 0.03	0.87 ± 0.02	1.16 ± 0.03	1.1 ± 0.04	1.06 ± 0.05	2.24 ± 0.11	1.75 ± 0.04 *	1.63 ± 0.11 *
Left ventricle weight (g)	0.62 ± 0.01	0.57 ± 0.02 *	0.55 ± 0.02 *	0.77 ± 0.02	0.69 ± 0.02 *	0.66 ± 0.02 *	1.3 ± 0.05	0.99 ± 0.03 *	0.88 ± 0.06 *
Right ventricle weight (g)	0.17 ± 0.01	0.17 ± 0.01	0.18 ± 0.01	0.24 ± 0.01	0.25 ± 0.01	0.23 ± 0.02	0.35 ± 0.01	0.33 ± 0.01	0.32 ± 0.02
Lung weight (g)	1.3 ± 0.02	1.29 ± 0.05	1.3 ±0.03	1.7 ± 0.12	2.98 ± 0.26 *	2.47 ± 0.29 *	1.64 ± 0.04	1.88 ± 0.32	1.50 ± 0.1

Values are presented as mean ± S.E.M., * *p* < 0.05 vs. control group, (*n* = 6–7, One-Way ANOVA, Holm-Sidak post hoc test). RT: radiotherapy only group (50 Gy), RT Los: RT plus losartan group.

**Table 2 ijms-22-12963-t002:** Post-RT blood cell counts, hemoglobin, and hematocrit values.

Parameter (Unit)	Week 1			Week 3			Week 15		
	Control	RT	RT Los	Control	RT	RT Los	Control	RT	RT Los
White blood cell counts (10^9^/L)	5.79 ± 0.43	6.18 ± 0.7	7.15 ± 0.36 *	4.44 ± 0.56	6.77 ± 0.8 *	7.16 ± 0.7 *	6.09 ± 0.58	6 ± 0.39	6.51 ± 0.52
Platelet counts (10^9^/L)	569 ± 8	587 ± 20	643 ± 33	559 ± 27	646 ± 35	673 ± 31 *	636 ± 17	586 ± 67	736 ± 53
Red blood cell counts (10^12^/L)	6.69 ± 0.11	7.37 ± 0.29 *	7.74 ± 0.2 *	7.54 ± 0.15	7.54 ± 0.18	7.95 ± 0.18	8.30 ± 0.15	9.16 ± 0.44	9.50 ± 0.3 *
Hemoglobin (g/L)	130 ± 3	143 ± 5	146 ± 3 *	143 ± 3	142 ± 3	152 ± 3	146 ± 2	165 ± 8 *	175 ± 7 *
Hematocrit (L/L)	0.41 ± 0.01	0.44 ± 0.02	0.45 ± 0.01	0.44 ± 0.01	0.45 ± 0.01	0.47 ± 0.01	0.43 ± 0.01	0.49 ± 0.02 *	0.50 ± 0.02 *

Values are presented as mean ± S.E.M., * *p* < 0.05 vs. control group, (*n* = 6–7, One-Way ANOVA, Holm-Sidak post hoc test). RT: radiotherapy only group (50 Gy), RT Los: RT plus losartan group.

**Table 3 ijms-22-12963-t003:** Echocardiographic parameters according to treatment groups at weeks 1, 3, and 15.

Parameter (Unit)	Week 1			Week 3			Week 15		
	Control	RT	RT Los	Control	RT	RT Los	Control	RT	RT Los
E velocity (m/s)	1.00 ± 0.04	0.94 ± 0.05	0.90 ± 0.04	1.00 ± 0.04	1.05 ± 0.03	0.93 ± 0.04 #	1.01 ± 0.02	0.81 ± 0.10 *	0.75 ± 0.05 *
e’ (m/s)	0.060 ± 0.002	0.053 ± 0.001 *	0.063 ± 0.005	0.066 ± 0.006	0.038 ± 0.002 *	0.056 ± 0.005 #	0.077 ± 0.007	0.035 ± 0.002 *	0.042 ± 0.002 *
SWTs (mm)	2.70 ± 0.10	2.84 ± 0.16	2.91 ± 0.10	3.14 ± 0.15	3.47 ± 0.12	3.18 ± 0.14	3.58 ± 0.12	4.64 ± 0.17 *	4.29 ± 0.32
SWTd (mm)	1.61 ± 0.10	1.60 ± 0.09	1.90 ± 0.06	1.79 ± 0.12	1.73 ± 0.06	1.77 ± 0.14	1.83 ± 0.04	3.23 ± 0.33*	3.12 ± 0.28
AWTs (mm)	2.64 ± 0.17	2.61 ± 0.09	2.92 ± 0.20	2.85 ± 0.17	3.19 ± 0.18	2.89 ± 0.27	3.3 ± 0.23	4.56 ± 0.28 *	4.51 ± 0.17 *
AWTd (mm)	1.63 ± 0.17	1.51 ± 0.07	1.75 ± 0.09	1.61 ± 0.07	1.67 ± 0.11	1.67 ± 0.09	1.99 ± 0.13	3.14 ± 0.31 *	3.04 ± 0.17 *
IWTs (mm)	2.24 ± 0.13	2.47 ± 0.08	2.70 ± 0.10 *	2.67 ± 0.07	3.32 ± 0.17 *	2.91 ± 0.23	3.44 ± 0.25	4.56 ± 0.09 *	3.69 ± 0.31
IWTd (mm)	1.48 ± 0.03	1.51 ± 0.09	1.67 ± 0.09	1.73 ± 0.07	1.98 ± 0.07 *	1.90 ± 0.15	2.01 ± 0.12	3.67 ± 0.12 *	2.72 ± 0.26 *#
LVEDD (mm)	7.73 ± 0.22	7.34 ± 0.27	6.98 ± 0.16 *	8.33 ± 0.2	7.4 ± 0.14 *	7.11 ± 0.29 *	8.44 ± 0.33	4.85 ± 0.17 *	5.67 ± 0.29 *#
LVESD (mm)	4.75 ± 0.21	4.05 ± 0.28	3.68 ± 0.21 *	4.83 ± 0.17	3.40 ± 0.20 *	3.25 ± 0.25 *	4.26 ± 0.55	0.98 ± 0.15 *	1.63 ± 0.16 *
FS (%)	39 ± 2	45 ± 2 *	48 ± 2 *	42 ± 2	54 ± 2 *	54 ± 2 *	50 ± 5	80 ± 3 *	71 ± 3 *
EF (%)	56 ± 2	57 ± 3	52 ± 1	53 ± 2	53 ± 3	53 ± 1	53 ± 2	57 ± 2	55 ± 1

Values are presented as mean ± S.E.M., * *p* < 0.05 vs. control group, # *p* < 0.05 vs. RT group (*n* = 6–7, One-Way ANOVA, Holm-Sidak post hoc test). RT: radiotherapy only group (50 Gy), RT Los: RT plus losartan group. AWT: anterior wall thickness, d: diastolic, E: early flow velocity, e’: velocity of the septal mitral annulus, EF: ejection fraction, FS: fractional shortening, IWT: inferior wall thickness, LVEDD: left ventricular end-diastolic diameter, LVESD: left ventricular end-diastolic diameter, s: systolic, SWT: septal wall thickness.

## Data Availability

The datasets used and/or analyzed during the current study are available from the corresponding authors on a reasonable request.

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
