# Peer review of "Investigation of the Antihypertrophic and Antifibrotic Effects of Losartan in a Rat Model of Radiation-Induced Heart Disease"

_ijms, 2021, doi:10.3390/ijms222312963_

Round 1

Reviewer 1 Report

The study by Kovacs et al., investigated losartan, an angiotensin-II receptor blocker on radiation induced heart disease. One major issue is: that the data presented did not support their conclusion: Losartan seems to alleviate left ventricular remodeling via inhibiting the overexpression of chymase and the TGF-ß/SMAD signaling pathway in our rat model of RIHD. 

Authors shall use inhibitors or shRNA of TGF-ß and SMAD to suppress the function of TGF-ß and SMAD. This can determine their role in losartan mediated function in radiation induced heart disease.

Several minor issues:

  1. EF is surprising high for animals in RT and RT+Los group at week-15. Authors shall discuss this. Authors shall discuss this in discussion part.
  2. For all western blot images, authors shall put all protein samples in one membrane instead cut and presented individually. Also author shall include a protein marker in each WB result. 

Reviewer 2 Report

In this article Kovács and colleagues show that administration of Losartan prevents myocardial dysfunction, alleviating left ventricular hypertrophy and cardiac fibrosis in a rat model of radiation-induced heart disease. There is various evidence that losartan, a selective antagonist of AT1 receptors for angiotensin II, decreases cardiac and skeletal muscle fibrosis and improves cardiac systolic function, but there are only 2 preclinical studies that explores the effects of angiotensin-II receptor blocker in cancer patient treated with thoracic radiotherapy.  The authors investigates the molecular mechanisms that could contribute to this beneficial effect. In fact, Losartan seems to inhibit the overexpression of chymase and the TGF-β/SMAD signaling pathway.

Therefore, I have some observations that needed to be addressed for the paper publication.

Minor points:

  • it might be interesting to evaluate hypertrophy with WGA (Wheat Germ Agglutinin Staining) and cardiac fibrosis with Masson's trichrome stain (Aquila I, Cianflone E, et al. c-kit Haploinsufficiency impairs adult cardiac stem cell growth, myogenicity and myocardial regeneration. Cell Death Dis. 2019 Jun 4;10(6):436. doi: 10.1038/s41419-019-1655-5)
  • the authors should quantify protein bands from western blot films by densitometry analisys.

Reviewer 3 Report

Kovacs et al wrote and interesting and generally well written article that is entitled: "Losartan alleviates left ventricular hypertrophy and cardiac fibrosis via inhibiting the TGF-β/SMAD signaling pathway in a rat model of radiation-induced heart disease."

I do have a couple of questions regarding this article: 

  • First and most importantly, I do wonder how established this rat model of RIHD is. I see that the authors provide citations of the model, but both appear from the same and own group. Is there any other group using this model? Please provide the citation.
  • Secondly, is this model able to be translated to humans / the clinical application?  A dose of 50 Gy to the whole heart seems extraordinarily high. Was this dose given in one fraction? What techniques were used to make sure that the beam did in fact target mostly the heart? What was the dose to the coronary arteries? 
  • This is probably similar to the second point: Which comparable doses that are used in humans? RIHD mostly develops after years in humans and not after weeks -- would advise to discuss this or even make a separate limitations section. 
  • Generally, I don't think that left ventricular hypertrophy is a response of the myocardium to radiation in humans. Does the rat myocardium respond differently in this regard? 
  • What did radiation do to the vascular bed? 

Round 2

Reviewer 1 Report

The paper was revised and improved. One minor issue, authors shall indicate in the original WB images from which and where the selected WB images were cut. 

Author Response

We thank the Reviewer for this comment. In the second revised supplementary material, colored frames indicate the places in the original WB images from where the selected representative bands were cut according to the Reviewer's request. Moreover, we changed the representative WB images of tERK1 and tERK2 at week 3 in Figures 7a and 7b to show images of the same samples in cases of pERK1/2 and tERK1/2 in the second revised MS. We hope that the modifications are satisfactory for the Reviewer now.

Reviewer 2 Report

The authors have clarified all the reviewers' requests and so I  consider that this manuscript can be accepted

Author Response

We thank the Reviewer for the positive feedback on our revised MS.

We modified the Supplementary material and Figure 7 according to the request of Reviewer 1 in the second revision.

Reviewer 3 Report

Thank you for addressing the reviewers comments. I do not have more to add. 

Author Response

(The authors gave the same response as above.)
